# The Effects of Walnuts and Academic Stress on Mental Health, General Well-Being and the Gut Microbiota in a Sample of University Students: A Randomised Clinical Trial

**DOI:** 10.3390/nu14224776

**Published:** 2022-11-11

**Authors:** Mauritz F. Herselman, Sheree Bailey, Permal Deo, Xin-Fu Zhou, Kate M. Gunn, Larisa Bobrovskaya

**Affiliations:** 1Health and Biomedical Innovation, Clinical and Health Sciences, University of South Australia, Adelaide, SA 5000, Australia; 2Allied Health & Human Performance, University of South Australia, Adelaide, SA 5000, Australia

**Keywords:** stress, depression, diet, gut-brain axis, microbiota, pathways, walnuts

## Abstract

Poorer mental health is common in undergraduate students due to academic stress. An interplay between stress and diet exists, with stress influencing food choices. Nutritional interventions may be effective in preventing mental health decline due to complex bidirectional interactions between the brain, the gut and the gut microbiota. Previous studies have shown walnut consumption has a positive effect on mental health. Here, using a randomized clinical trial (Australian New Zealand Clinical Trials Registry, #ACTRN12619000972123), we aimed to investigate the effects of academic stress and daily walnut consumption in university students on mental health, biochemical markers of general health, and the gut microbiota. We found academic stress had a negative impact on self-reported mood and mental health status, while daily walnut consumption improved mental health indicators and protected against some of the negative effects of academic stress on metabolic and stress biomarkers. Academic stress was associated with lower gut microbial diversity in females, which was improved by walnut consumption. The effects of academic stress or walnut consumption in male participants could not be established due to small numbers of participants. Thus, walnut consumption may have a protective effect against some of the negative impacts of academic stress, however sex-dependent mechanisms require further study.

## 1. Introduction

While humans have evolved mechanisms to cope with acute stressful events, modern life presents many challenges which can lead to chronic stress, which we are less well-adapted to, potentially precipitating into psychiatric disorders such as anxiety and depression. It can also contribute to certain metabolic disorders such as diabetes [1]. Many studies have highlighted the growing problem of poor mental health in undergraduate students as a result of the chronic nature of academic stress and a recent World Health Organisation Mental Health Survey found at least 75% of mental health disorders occur prior to 24 years of age, making undergraduate students particularly vulnerable to depression [2,3,4,5]. The association between chronic stress and depression may be due to the overuse or dysregulation of the stress response systems, namely the hypothalamic-pituitary-adrenal (HPA) axis and sympatho-adrenomedullary system [6,7,8]. Dysregulation of the former system results in chronically elevated levels of cortisol, known to impair learning and memory via key brain regions such as the hippocampus, basolateral amygdala and medial prefrontal cortex, with overactivation of the latter system leading to increased output of catecholamines [6,7,8]. In the brain, chronic stress can perturb the mesolimbic dopaminergic system which can lead to decreased reward sensitivity, a common feature of depression symptomatology [9]. Previous studies have found that salivary α-amylase may be useful as a marker of activation of the sympatho-adrenomedullary system, since it is closely associated with catecholamine levels during periods of stress, and higher levels exist in patients with major depression compared with healthy individuals [6,10]. Furthermore, these perturbations in learning, memory and reward systems, driven by dysregulated stress responses, are thought to be a result of altered neuroplasticity and reduced expression of the mature form of brain-derived neurotrophic factor (mBDNF) in the brain [7].

There is a relationship between stress and diet, with stress not only influencing food choices, but also altering metabolic responses to food [11]. Recent research has also shown that nutritional interventions such as the Mediterranean diet are effective in preventing psychological disorders such as depression [12], and that nut consumption may lower depression risk [13]. Nutritional interventions may be effective in this way due to the complex bidirectional interactions between the brain, the gut and the gut microbiota (readers are referred to a previous review [14]). Acting together, stress and diet are thought to modulate gut health and the composition of the gut microbiota, although the gut microbiota may also exert influence over the brain and the gut via the production of metabolites, further influencing food cravings and mood [11].

Evidence suggests that walnuts may have positive effects on cognition and mental health, owing to their abundance in omega-3 fatty acids, α-linolenic acid and tryptophan content [15,16,17]. Furthermore, walnuts are a rich source of several neuro-supportive compounds such as melatonin, polyphenols, folate and vitamin E [18]. According to a recent systematic review of the effects of nut consumption on the human gut microbiota, daily walnut consumption for 8 weeks was sufficient to result in a shift in β-diversity, but not α-diversity in faecal samples, the former being a measure of dissimilarity between samples and the latter a measure of richness within each sample [19]. The review highlighted that several studies identified decreases in the relative abundance of the Actinobacteria phylum, indicating that daily walnut consumption can have profound effects on the gut microbiota, likely due to their nutritional composition [19].

This study aimed to firstly, evaluate the relationship between academic stress, as a chronic stressor, and mental health, biochemical markers of general health, and the diversity and composition of the gut microbiota, and secondly, to evaluate the effects of daily walnut consumption on these parameters in a sample of university students.

## 2. Materials and Methods

### 2.1. Participants

Healthy male and female undergraduate students aged between 18 and 35 were recruited through advertisements on campus at the University of South Australia, Australia. All participants were screened prior to enrolment into the study by an online screening questionnaire, followed by an informational group meeting and a diet and lifestyle questionnaire. Individuals who consumed >2 serves (56 g) of nuts per week or >2 capsules of fish oil per day, used any medication for depression, anxiety or any other neurological or psychiatric conditions, or used oral or pulmonary steroids in the previous 3 months were excluded from the study. Any individuals in cohorts 2 & 3 (see Section 2.2) who used any antibiotics or probiotic supplements in the previous month were excluded from the study to account for the inclusion of faecal microbiota analyses (see Section 2.5). Individuals with any nut allergies, who were night shift workers or who were pregnant were also excluded. All participants were asked to maintain their level of physical activity throughout the study. Written informed consent was obtained from all participants and the study was approved by the institutional review board of the University of South Australia, South Australia, Australia. Of 214 expressions of interest, a total of 80 eligible participants were included in the study and randomised, using simple randomisation, to either the treatment group or control group.

### 2.2. Study Design

The study was a randomised feeding trial and is registered on the Australian New Zealand Clinical Trials Registry (Registration Number ACTRN12619000972123). The trial took place during the university semester where participants were either allocated to a treatment group where they were provided with fresh pre-portioned walnuts and asked to consume 1 portion (approximately 56 g) per day for the 16-week duration of the study, while the control group were asked to refrain from consuming any type of nut or fatty fish for the same duration. Participants were allowed to incorporate each portion of walnuts into their existing diet at any time of the day. Previous studies have administered walnuts to their participants in amounts between 30 g and 75 g [20,21] although the currently recommended intake in Australian nutritional guidelines is about 30 g [22]. The portion of 56 g of walnuts administered to participants was chosen based on the previously published literature and work of Pribis et al. [17] who administered 60 g of walnuts to university students. A 16-week duration for walnut consumption was based on the timeline of the University of South Australia’s teaching semester and examination period (Figure 1). Recruitment and clinical visits were conducted across 3 university semesters within a 2-year period, making up 3 cohorts of participants. Cohort 1 was conducted in semester 2 2019 (August–December), cohort 2 in semester 2 2020 (August–December) and cohort 3 in semester 1 2021 (February–July). All participants attended 3 clinical visits (Figure 1) at the University of South Australia Clinical Trial Facility where a blood sample and saliva sample were provided at each clinical visit. The first clinical visit served as a baseline at the beginning of the university semester, with the second visit occurring during the examination period and the final visit 2 weeks following. All participants completed a series of questionnaires at each clinical visit including the 21-item Depression Anxiety Stress Scale (DASS21), Perceived Stress Scale (PSS), Short-form Mental Health Continuum Questionnaire (MHC), Profile of Mood States Questionnaire (POMS), 8-dimension Assessing Quality of Life Questionnaire (AQoL-8D), the Active Australia Survey (AAS) and the Leed’s Sleep Evaluation Questionnaire (LSE). The POMS was defined as a primary outcome measure amongst the questionnaires, while all other questionnaires were defined as secondary outcome measures. 

The participants of the study were made up of 3 separate cohorts due to the study utilising the university semester as a timeline. As a result, cohorts were spread across different university semesters with cohort 1 and cohort 2 in the second half of the year in 2019 and 2020, respectively, and cohort 3 in the first half of the year in 2021. Participants in cohort 2 and cohort 3 were also given the option to provide a faecal sample at each clinical visit. It should also be disclosed that cohort 2 attended clinical visits during the 2020 COVID-19 pandemic which could be considered a confounder. Overall, 80 participants were enrolled into the study across the 3 cohorts (Figure 2).

### 2.3. Blood Sample Collection and Analyses

Non-fasting blood samples were obtained at each clinical visit. Blood was collected between the hours of 12:00 p.m. and 4:00 p.m. into dipotassium ethylenediaminetetraacetic acid (K2 EDTA) coated Vacuette tubes (Greiner BioOne, Kremsmünster, Austria) to obtain plasma and into blood clotting activator & thrombin (CAT) serum clot activator Vacuette tubes (Griener BioOne, Kremsmünster, Austria) for serum. Plasma samples were immediately placed on ice and centrifuged within 30 min of collection at 4 °C at 4000× *g* RPM for 15 min. Serum samples were allowed to separate for 45 min prior to centrifugation at 4 °C at 4000× *g* RPM for 15 min. Plasma and serum samples were stored immediately at −80 °C following centrifugation. Total phenolic content, the ferric reducing antioxidant potential and the oxygen radical absorbance capacity of plasma were determined as described elsewhere [23]. Serum triglycerides, cholesterol, glucose, total protein, albumin, total bilirubin, direct bilirubin, indirect bilirubin and C-reactive protein were measured using the automated Indiko Plus Clinical Chemistry Analyser (Catalog no. 98640000, Thermo Fisher Scientific, Adelaide, Australia). As a primary outcome measure, plasma cortisol was measured using a commercial ELISA kit (Catalog no. DEH3388, Demeditec Diagnostics GmbH, Kiel, Germany). Serum mBDNF was measured using an in-house mBDNF ELISA assay [24].

### 2.4. Saliva Sample Collection and Analyses

Saliva samples were collected from participants at each clinical visit. Participants were asked to refrain from eating, drinking, taking any medications or performing any oral hygiene procedures for at least 1 h prior to collection. Participants were also asked to refrain from consuming any alcohol at least 12 h prior to their clinical visit. Participants were provided with water to rinse out their mouth 10 min prior to collection. Saliva was collected via a synthetic swab (Catalog no. 51.1534, Sarstedt Australia, Adelaide, Australia) placed underneath the participant’s tongue for a minimum duration of 2 min until fully saturated with saliva. Saliva samples were placed on ice immediately and centrifuged within 1 h of collection at 4 °C at 1780× *g* RPM for 10 min and immediately stored at −80 °C. Salivary α-amylase was measured using a commercial enzymatic colorimetric assay (Catalog no. RE80111, IBL International, Hamburg, Germany).

### 2.5. Faecal Sample Collection 16S rRNA Gene Sequencing

As a secondary outcome measure, faecal samples were collected by participants in cohort 2 and 3 at each clinical visit to assess the composition of the gut microbiota. Results from cohort 2 and 3 were pooled (Females: control *n* = 14, walnut *n* = 11; Males: control *n* = 3, walnut *n* = 5). Participants were provided with faecal sample collection kits containing DNA stabilisation buffer and instructed to collect samples according to the manufacturer’s instructions (Catalog no. SC010, SC011, SC012, Canvax Biotech, Córdoba, Spain). Samples were stored at −80 °C, and microbial DNA extraction, amplification, sequencing and diversity profiling was performed by the Australian Genomics Research Facility (AGRF, Adelaide, Australia). Briefly, the 16S rRNA genes (V3–V4) were amplified using 341F (CCTAYGGGRBGCASCAG) and 806R (GGACTACNNGGGTATCTAAT) primers. Next Generation Illumina sequencing was employed for amplicon sequencing of the DNA library by AGRF, and 300-bp paired-end reads were generated. Primary image analysis was performed using MiSeq Control Software v3.1.0.12 and Real Time Analysis v1.18.54.4. Diversity profiling was performed with Quantitative Insights into Microbial Ecology (QIIME 2 2019.7) [25]. The demultiplexed raw reads were primer trimmed and quality filtered using the cutadapt plugin followed by denoising with DADA2 by AGRF [26]. Taxonomy was assigned to amplicon sequence variants using the q2-feature-classifier classify-sklearn naïve Bayes taxonomy classifier. The SILVA rRNA database [27,28] was used to annotate taxonomic information for the representative sequences of the OTUs.

### 2.6. Microbiota Analysis

The sequenced 16S rRNA data provided by AGRF were further analysed using QIIME 2 (2022.2). These data were rarefied to a depth of 19,300 sequences prior to analysis of α- and β-diversity to ensure even sampling with 100% retention of samples. The Shannon α-diversity index was used to determine species richness and evenness within each sample. The Bray–Curtis β-diversity index was used to determine dissimilarities between microbial populations between samples. Differences in relative abundance were evaluated at the phylum level to identify major shifts in the gut microbiome. Following Analysis of Compositions of Microbiomes with Bias Correction (ANCOM) analysis (analysis not shown) using QIIME 2 (2022.2), differences in relative abundance were evaluated at the genus level. 

### 2.7. Statistical Analyses

Sample size of 44 participants per group was estimated using G*Power (3.1.9.2) using a priori repeated measures ANOVA with an effect size and power set to 25% and 95%, respectively. Effect size was estimated based on the primary outcome measure of the study to detect at least a 25% difference in mood scores in the POMS. The outbreak of COVID-19 during 2020 prevented the recruitment of 44 participants/group and contributed to a participant withdrawal rate that was higher than anticipated. Baseline characteristics between groups were assessed using unpaired *t*-tests and chi-squared tests for continuous and categorical variables, respectively. Mixed effects analyses were carried out using PRISM v8.3 (GraphPad Software, Inc., San Diego, CA, USA) with treatment as a between subject factor and time as a repeated measurement. The Shapiro–Wilk normality test was used to determine the distribution of the data prior to mixed effects analysis. Post hoc multiple comparisons were made with Bonferroni adjustments. Pearson’s correlation analyses were used to analyse associations between the data. The α-diversity Shannon index and β-diversity index were compared between the control and treatment group, considering each clinical visit, using mixed effects analyses and PERMANOVA tests, respectively. Statistical significance was set at *p* < 0.05.

## 3. Results

### 3.1. Baseline Characteristics

At four weeks into the university semester (see Section 2.2, Figure 1) baseline characteristics of the study population (Table 1) were assessed showing the study participants in each group had a mean age of 22 years, with the control group having a mean body mass index (BMI) of 22.86 ± 0.83 $\mathrm{kg}/m^{2}$ and the treatment group had a similar mean BMI of 22.82 ± 0.73 $\mathrm{kg}/m^{2}$. Most participants were female (75%), however there was no significant disproportion of sex in the control group versus the treatment group ($X^{2}$(1, *N* = 60) = 0.8, *p* = 0.3711). Most of the study participants were of Asian ethnicity (55%), followed by Caucasian, African, Afghani, Persian and South American ethnicities. More participants had part-time employment (57%) and just under half the number of participants had 4 examinable subjects (45%), followed by 3 (25%), 2 (25%) and 1 (5%) throughout the university semester. Participants who only had 1 examinable subject also had at least 1 additional major university assignment due during examination week.

### 3.2. Questionnaires

Mental health, mood, general well-being and sleep habits were evaluated across 3 clinical visits during a university semester, using a series of questionnaires: the DASS21 [29], the PSS [30], the MHC-SF [31], the POMS questionnaire [32], the AQoL-8D questionnaire [33] and the LSE questionnaire [34].

The mean scores for the DASS21 are presented in Figure 1. Responses to the DASS21 are divided up into 3 subscales: Depression, Anxiety and Stress, each assessing participants feelings and thoughts that occurred during the past month. For the Depression subscale, a significant increase in score was observed in the control group at visit 2 which coincided with the university examination period (4.691 ± 1.02; *p* = 0.0002), with a significant decrease from visit 2 to visit 3 after the examination period (−3.692 ± 1.24; *p* = 0.0181). No such increase was observed at visit 2 in the walnut group; however, a significant decrease was observed at visit 3 (−3.103 ± 0.98; *p* = 0.0113). Mixed effects analyses revealed a significant main effect of time (*p* = 0.0002) with depression scores increasing from visit 1 to visit 2 (*p* = 0.0193) and decreasing from visit 2 to visit 3 (*p* = 0.0002) indicating that examination stress increased perceived measures of depression. Walnut consumption appeared to have a positive effect on these measures with a near significant interaction effect between time and treatment (*p* = 0.0519).

Scores for the Anxiety subscale revealed no significant differences between the treatment and control groups; however, a significant main effect of time was evident (*p* = 0.0053) with anxiety scores increasing from visit 2 to visit 3 (*p* = 0.0051). Meanwhile, the Stress subscale scores revealed a significant score increase in the control group at visit 2 (5.333 ± 1.20; *p* = 0.0003), with no such increase observed in the treatment group. The treatment group showed a significant score decrease at visit 3 (−2.828 ± 0.77; *p* = 0.0031). Overall, there was a significant main effect of time (*p* < 0.0001) with stress scores increasing between visit 1 and 2 (*p* = 0.0002) and decreasing between visit 2 and 3 (*p* = 0.0008), suggesting that the university examination period increased perceived measures of stress in participants. While the treatment group did not appear to show such trends, no significant interaction effect between time and walnut consumption was found.

The mean scores for the PSS are presented in Figure 2. The PSS is given a total score which is made up of a positive subscale made up of reversed item scores and negative subscale, with higher scores indicating greater distress [35]. Each participant was asked to respond based on their thoughts during the past month. While no significant changes were observed for the total score and positive subscale for either of the participant groups, a significant score increase was observed in the negative subscale in the control group at visit 2 (1.612 ± 0.59; *p* = 0.0327), with a significant main effect of time (*p* = 0.0184) showing an overall decrease between visit 2 and visit 3 (*p* = 0.0253), indicating that some distress was perceived in the control group at visit 2. No such changes were observed in the treatment group.

The mean scores for the MHC are presented in Figure 3. The MHC comprises 14 items, with 3 items assessing Emotional Well-Being, 5 items assessing Social Well-Being and 6 items assessing Psychological Well-Being, overall making up 3 dimensions with higher scores representing better overall mental health [31]. All participants were asked to respond based on how they felt in the past month. While no significant differences were observed in both the treatment and control groups across the university semester for the Emotional Well-Being and Social Well-Being dimensions, a significant decrease was observed in the Psychological Well-Being dimension in the control group at visit 2 compared to visit 1 (−3.242 ± 0.85; *p* = 0.0021) as well as at visit 3 compared to visit 1 (−3.518 ± 0.85; *p* = 0.0105), with no such effect observed in the treatment group. Mixed effect analyses revealed a significant main effect of time (*p* = 0.0003) with scores decreasing between visit 1 and visit 2 (*p* = 0.0002) as well as between visit 2 and visit 3 (*p* = 0.001). Overall, these results indicate that the psychological well-being of the control participants was negatively impacted by academic stress at visit 2, while those in the treatment group did not experience this effect.

The mean scores for the POMS are presented in Figure 4. The POMS assesses distinct mood states across 5 dimensions (Anger-Hostility; Confusion-Bewilderment; Depression-Dejection; Fatigue-Inertia; Tension-Anxiety) using items on a 5-point scale [36]. All participants were asked to respond to the questionnaire based on how they felt at present. Scores from each of the 5 dimensions were summed to provide a total mood disturbance score. A significant increase in total mood disturbance was observed in both the treatment and control group at visit 2 compared to visit 1 (Control: 14.09 ± 4.46; *p* = 0.011. Walnut: 7.567 ± 2.94; *p* = 0.0472), with this score decreasing for both groups again at visit 3 (Control: −15.63 ± 5.24; *p* = 0.0176. Walnut: −14.26 ± 2.51; *p* < 0.0001). A significant main effect of time on total mood disturbance was also observed (*p* < 0.0001) with increased scores between visit 1 and visit 2 (*p* = 0.0004) and decreased between visit 2 and visit 3 (*p* < 0.0001), indicating overall that academic stress experienced at visit 2 had a negative effect on mood in participants. Regarding each of the dimensions, a significant main effect of time on each dimension score for all dimensions was observed (Anger-Hostility: *p* = 0.0115; Confusion-Bewilderment: *p* < 0.0001; Depression-Dejection: *p* = 0.0223; Fatigue-Inertia: *p* < 0.0001; Tension-Anxiety: *p* < 0.0001), with only the Fatigue-Inertia and Tension-Anxiety dimensions showing a significant overall increase on score from visit 1 to visit 2 (*p* < 0.0001; *p* < 0.0001). No significant differences were found for Anger-Hostility scores at visit 2, while a significant decrease was observed at visit 3 for the treatment group compared to their scores at visit 2 (−1.324 ± 0.43; *p* = 0.0138). A significant increase was observed for Confusion-Bewilderment, Fatigue-Inertia and Tension-Anxiety scores in the control group at visit 2 compared to visit 1 (Confusion-Bewilderment: 1.491 ± 0.58; *p* = 0.0488. Fatigue-Inertia: 4.239 ± 0.0009. Tension-Anxiety: 3.397 ± 1.07; *p* =0.0102) as well as decreases at visit 3 compared to visit 2 (Confusion-Bewilderment: −1.998 ± 0.59; *p* = 0.0064. Fatigue-Inertia: −4.036 ± 1.05; *p* = 0.002. Tension-Anxiety: −4.941 ± 1.31; *p* = 0.0023), indicating that these dimensions contributed to the increased total mood disturbance observed at visit 2. While walnut consumption did not attenuate increases in scores for any of the dimensions, there were significant decreases in scores from visit 2 to visit 3 (Anger-Hostility: −1.324 ± 0.43; *p* = 0.0138. Confusion-Bewilderment: −2.562 ± 0.68; *p* = 0.0022. Fatigue-Inertia: −4.209 ± 0.78; *p* < 0.0001. Tension-Anxiety: −4.099 ± 0.87; *p* = 0.0002).

The mean scores for the AQoL-8D are presented in Figure 5. The AQoL-8D comprises two super-dimension components: Physical which includes the dimensions of independent living, senses and pain; and Psychosocial which includes the dimensions of mental health, relationships, coping, self-worth and happiness [37]. All participants were asked to respond based on their experiences in the past week. Only remarkable results from the Psychosocial dimensions have been presented (all results are available in Appendix A). A significant decrease in Overall Quality of Life score (obtained from the sum of all individual dimension scores) was observed at visit 2 compared to visit 1 for the control group (−3.244 ± 1.18; *p* = 0.0297), however this did not occur in the treatment group at the same interval. The score for Overall Quality of Life increased at visit 3 compared to visit 2 for both the treatment and control groups (Control: 3.543 ± 1.07; *p* = 0.0079. Walnut: 3.6 ± 0.74; *p* = 0.0001). For the individual dimensions, the dimensions for coping showed a significant decrease for controls at visit 2 (−8.023 ± 2.55; *p* = 0.0115) and an increase for both the control and treatment groups at visit 3 (Control: 6.197 ± 2.41; *p* 0.0467. Walnut: 4.673 ± 1.34; *p* = 0.0047). The dimension for mental health showed a significant decrease for controls at visit 2 (−7.172 ± 2.26; *p* = 0.0107) and an increase for both the controls and treatment groups at visit 3 (8.478 ± 1.88; *p* = 0.0003). These results suggest that academic stress experienced at visit 2 increased perceived feelings of sadness and worry, and decreased perceptions of being able to cope with problems. These effects were not observed with daily walnut consumption.

The mean scores for the LSE questionnaire are presented in Figure 6. The LSE questionnaire assesses changes in sleep quality over time through self-reporting across 10-items which evaluates the 4 domains of ease of getting to sleep, quality of sleep, ease of waking following sleep and behaviour following wakening [38]. Each participant was asked to rate items based on the sleep which they experienced in the previous night. Higher scores on items indicate a poorer outcome. Across the study period, a main effect of time was observed on all four dimensions of the LSEQ (Getting to Sleep: *p* = 0.0013; Quality of Sleep: *p* = 0.0087; Awake following Sleep: *p* = 0.0442; Behaviour following Wakening: *p* = 0.0055). A significant main effect of walnut consumption versus a nut-free diet was also observed in the Getting to Sleep dimension (*p* = 0.0013). Compared to visit 2, walnut supplementation decreased scores related to getting to sleep (−42.38 ± 9.95; *p* = 0.0007), quality of sleep (−32.80 ± 11.09; *p* = 0.0191), ease of wakefulness (−24.60 ± 7.382; *p* = 0.0078), and behaviour following sleep (27.6 ± 8.025; *p* = 0.0049) by visit 3, however no significant decreases were observed in scores for the control group. Comparing scores from visit 3 to visit 1 also revealed walnut supplementation decreased scores related to behaviour following sleep (−26.35 ± 5.98; *p* = 0.0003).

Following analysis of the questionnaires, biochemical analyses were carried out on blood samples collected at each clinical visit.

### 3.3. Biomarkers of General Health

Total polyphenol content, ferric reducing ability and oxygen radical absorbance capacity of plasma was evaluated in participants (Appendix A). Walnut supplementation did not increase the total polyphenol content of plasma nor was it significantly altered in either of the groups throughout the study. Similarly, plasma ferric reducing ability and oxygen radical absorbance capacity were not affected by walnut consumption nor academic stress.

Differences in serum biomarkers of general health were assessed in all participants to evaluate the biochemical effects of walnut consumption (Table 2). Total triglycerides and cholesterol levels were unaffected by walnut consumption, although total cholesterol levels were decreased during and at the end of the semester (visit 2 and 3) compared to the beginning of the semester (visit 1) (*p* = 0.0202), suggesting that cholesterol decreased in all participants through the course of the university semester. Serum glucose levels were unaffected by both academic stress and walnut consumption. There was a significant increase in total protein levels (*p* = 0.0116) and albumin levels (*p* = 0.0016) with walnut consumption. A significant main effect of time was also evident for total protein levels (*p* = 0.0204) indicating that these were decreased by academic stress, since a significant decrease in levels was found in the control group in visit 3 compared to visit 1 (*p* = 0.0193). Walnut consumption appeared to alleviate this decrease although no interaction effect was found (*p* = 0.3740). 

Regarding biomarkers of liver function, levels of total bilirubin, direct bilirubin and indirect bilirubin were unaffected by walnut consumption, although an increasing trend in total bilirubin and indirect bilirubin was observed in participants regardless of their group with a near-significant main effect of time (*p* = 0.0505 and *p* = 0.0729), suggesting that academic stress may lead to a mild increase in total bilirubin levels. C-reactive protein (CRP), a biomarker of inflammation produced by the liver, showed no significant differences between the groups at any visit, indicating that neither academic stress nor walnut consumption had any influence over inflammation in the participants.

Salivary α-amylase and plasma cortisol were assessed as biomarkers of stress across the university semester (Figure 7). A significant main effect of treatment was found for salivary α-amylase (*p* = 0.0305) and a significant decrease in α-amylase levels were evident at visit 2 in participants who received walnut intervention compared with controls (−56.10 ± 20.89; *p* = 0.0322). No changes to salivary α-amylase levels were found in controls at any of the clinical visits. Similarly, analysis of plasma cortisol showed no significant differences in both the treatment and control group across the university semester. Overall, these results suggest that while daily dietary intervention with walnuts appears to dampen release of salivary α-amylase, academic stress did not increase these biomarkers of stress.

Levels of the mature form of brain-derived neurotrophic factor (mBDNF) were assessed in the serum of all participants (Figure 8). Compared to visit 1, serum mBDNF was significantly decreased in the control group at visit 2 (−2803 ± 800.90; *p* = 0.0058) and at visit 3 (−5197 ± 1022; *p* = 0.0001), while levels were decreased in the walnut group at visit 3 compared to both visit 1 (−4776 ± 1090; *p* = 0.0006) and visit 2 (2638 ± 810.6; *p* = 0.0109). A significant main effect of time was also found (*p* < 0.0001). Overall, these results suggest that academic stress across the university semester is sufficient to decrease serum mBDNF levels and, as shown at visit 3, this effect is sustained following removal of academic stress. 

The results from the questionnaires and biochemical markers demonstrated effects of both academic stress and walnut consumption on certain parameters, thus, the relationships between these parameters were explored.

### 3.4. Correlation Analyses

During clinical visit 1 (Figure 9A), which occurred during the first half of the semester, no significant correlations were evident for albumin, aside from a positive correlation with total protein as expected. Significant positive correlations were identified between total protein and total cholesterol ($r^{2}$ = 0.288; *p* = 0.030), between total protein and triglycerides ($r^{2}$ = 0.313; *p* = 0.018), and a near significant negative correlation between total protein and the DASS21 Depression. Near significant negative correlations existed between total cholesterol and mBDNF, and between total cholesterol and Depression and Stress scores for the DASS21. A significant negative correlation was found at visit 1 between total cholesterol and the DASS21 Anxiety score ($r^{2}$ = −0.265; *p* = 0.046). CRP was significantly positively correlated with triglycerides ($r^{2}$ = 0.321; *p* = 0.015), glucose ($r^{2}$ = 0.368; *p* = 0.005), and Quality of Sleep in the LSE ($r^{2}$ = 0.394; *p* = 0.002), while CRP was negatively correlated with total bilirubin ($r^{2}$ = −0.278; *p* = 0.036), indirect bilirubin ($r^{2}$ = −0.343; *p* = 0.009), and Quality of Life scores in the AQoL-8D ($r^{2}$ = −0.281; *p* = 0.034). Cortisol showed a negative correlation with direct bilirubin ($r^{2}$ = −0.300; *p* = 0.038) and Total Mood Disturbance scores in the POMS ($r^{2}$ = −0.300; *p* = 0.041). Near significant correlations were also found between cortisol and total bilirubin, indirect bilirubin and the Psychological Well-being dimension of the MHC. α-Amylase, another biomarker of the stress response, showed a near significant correlation with the Social Well-being dimension of the MHC. Amongst the questionnaires, the DASS21 Depression subscale score was positively correlated with both the Anxiety subscale score ($r^{2}$ = 0.527; *p* = 0.000) and Stress subscale score ($r^{2}$ = 0.608; *p* = 0.000). The Quality of Sleep score from the LSE was positively correlated with the scores for Depression ($r^{2}$ = 0.335; *p* = 0.011), Anxiety ($r^{2}$ = 0.517; *p* = 0.000) and Stress ($r^{2}$ = 0.459; *p* = 0.000) in the DASS21. The PSS Total score was negatively correlated with the Emotional Well-being ($r^{2}$ = −0.473; *p* = 0.000), Social Well-being ($r^{2}$ = −0.432; *p* = 0.001) and Psychological Well-being ($r^{2}$ = −0.553; *p* = 0.000) dimensions in the MHC. The three dimensions of the MHC were all positively correlated with each other (Emotional Well-being and Social Well-being: $r^{2}$ = 0.628; *p* = 0.000; Emotional Well-being and Psychological Well-being: $r^{2}$ = 0.735; *p* = 0.000; Social Well-being and Psychological Well-being: $r^{2}$ = 0.743; *p* = 0.000). Total Mood Disturbance scores in the POMS were negatively correlated with the Emotion Well-being ($r^{2}$ = −0.620; *p* = 0.000), Social Well-being ($r^{2}$ = −0.546; *p* = 0.000) and Psychological Well-being ($r^{2}$ = −0.647; *p* = 0.000) dimensions of the MHC. Total Mood Disturbance in the POMS was also negatively correlated with Quality of Life scores in the AQoL-8D ($r^{2}$ = −0.332; *p* = 0.012). While Quality of Life scores in the AQoL-8D were positively correlated with the Emotional Well-being ($r^{2}$ = 0.308; *p* = 0.020) and Social Well-being ($r^{2}$ = 0.264; *p* = 0.047) dimensions of the MHC.

During clinical visit 2 (Figure 9B), which occurred during the university examination period, no significant correlations were evident for albumin, aside from a positive correlation with total protein as expected. Near significant correlations were found between total protein and total cholesterol, total protein and the DASS21 Stress score, α-amylase and triglycerides, α-amylase and glucose, α-amylase and total bilirubin as well as indirect bilirubin. Near significant correlations were also found between mBDNF and glucose, total bilirubin and the Social Well-being dimension of the MHC, indirect bilirubin and the Social Well-being dimension of the MHC, indirect bilirubin and Total Mood Disturbance in the POMS, and between the total score of the PSS and quality of sleep in the LSE.

Significant correlations were found at visit 2 (Figure 9B) for total protein and α-amylase ($r^{2}$ = −0.312; *p* = 0.037), total protein and the Emotional Well-being dimension of the MHC ($r^{2}$ = −0.344; *p* = 0.018), total protein and the Overall Quality of Life dimension of the AQoL8D ($r^{2}$ = −0.332; *p* = 0.021), total cholesterol and α-amylase ($r^{2}$ = −0.366; *p* = 0.011), total cholesterol and the DASS21 Depression score ($r^{2}$ = −0.317; *p* = 0.020), and total cholesterol and the DASS21 Stress score ($r^{2}$ = −0.337; *p* = 0.012). Cortisol was also significantly correlated with total bilirubin ($r^{2}$ = −0.413; *p* = 0.003), direct bilirubin ($r^{2}$ = −0.413; *p* = 0.003) and indirect bilirubin ($r^{2}$ = −0.397; *p* = 0.005), while CRP was correlated with the total score in the PSS ($r^{2}$ = −0.350; *p* = 0.017). The questionnaires also showed significant correlations between the DASS21 Depression score and the DASS21 Anxiety score ($r^{2}$ = 0.420; *p* = 0.001), the DASS21 Depression score and the DASS21 Stress score ($r^{2}$ = 0.756; *p* = 0.000), the DASS21 Anxiety score and the DASS21 Stress score ($r^{2}$ = 0.635; *p* = 0.000), the Emotional Well-being and the Social Well-being dimensions of the MHC ($r^{2}$ = 0.735; *p* = 0.000), the Emotional Well-being and Psychological Well-being dimensions of the MHC ($r^{2}$ = 0.756; *p* = 0.000), and the Social Well-being and Psychological Well-being dimensions of the MHC ($r^{2}$ = 0.747; *p* = 0.000). Total Mood Disturbance in the POMS was also significantly correlated with the Emotional ($r^{2}$ = −0.568; *p* = 0.000), Social ($r^{2}$ = −0.573; *p* = 0.000) and Psychological ($r^{2}$ = −0.451; *p* = 0.001) Well-being dimensions of the MHC, while Overall Quality of Life in the AQoL was significantly correlated with the Emotional ($r^{2}$ = 0.497; *p* = 0.000) and Psychological ($r^{2}$ = 0.401; *p* = 0.005) Well-being dimensions of the MHC and with Total Mood Disturbance in the POMS ($r^{2}$ = −0.375; *p* = 0.009).

By clinical visit 3 (Figure 9C), a prominent overall shift in parameters occurred. Total protein showed positive correlations with albumin ($r^{2}$ = 0.879; *p* = 0.000), total cholesterol ($r^{2}$ = 0.809; *p* = 0.000), triglycerides ($r^{2}$ = 0.418; *p* = 0.001), glucose ($r^{2}$ = 0.828; *p* = 0.000), total bilirubin ($r^{2}$ = 0.521; *p* = 0.000), direct bilirubin ($r^{2}$ = 0.639; *p* = 0.000), indirect bilirubin ($r^{2}$ = 0.456; *p* = 0.000), cortisol ($r^{2}$ = 0.547; *p* = 0.000), and mBDNF ($r^{2}$ = 0.658; *p* = 0.000). Albumin was positively correlated with total cholesterol ($r^{2}$ = 0.890; *p* = 0.000), triglycerides ($r^{2}$ = 0.426; *p* = 0.001), glucose ($r^{2}$ = 0.961; *p* = 0.000), total bilirubin ($r^{2}$ = 0.597; *p* = 0.000), direct bilirubin ($r^{2}$ = 0.740; *p* = 0.000), indirect bilirubin ($r^{2}$ = 0.520; *p* = 0.000), cortisol (*r*^2^= 0.600; *p* = 0.000) and mBDNF ($r^{2}$ = 0.700; *p* = 0.000). Total cholesterol was positively correlated with triglycerides ($r^{2}$ = 0.448; *p* = 0.000), glucose ($r^{2}$ = 0.865; *p* = 0.000), total bilirubin ($r^{2}$ = 0.571; *p* = 0.000), direct bilirubin ($r^{2}$ = 0.619; *p* = 0.000), indirect bilirubin ($r^{2}$ = 0.530; *p* = 0.000), cortisol ($r^{2}$ = 0.570; *p* = 0.000), mBDNF ($r^{2}$ = 0.596; *p* = 0.000) and negatively correlated with the Social Well-being dimension of the MHC ($r^{2}$ = −0.262; *p* = 0.049). Triglycerides were positively correlated with glucose ($r^{2}$ = 0.374; *p* = 0.004) and CRP ($r^{2}$ = 0.371; *p* = 0.005), while glucose was positively correlated with total bilirubin ($r^{2}$ = 0.574; *p* = 0.000), direct bilirubin ($r^{2}$ = 0.716; *p* = 0.000), indirect bilirubin ($r^{2}$ = 0.498; *p* = 0.000), cortisol ($r^{2}$ = 0.589; *p* = 0.000) and mBDNF ($r^{2}$ = 0.666; *p* = 0.000). Total bilirubin was positively correlated with direct bilirubin ($r^{2}$ = 0.924; *p* = 0.000), indirect bilirubin ($r^{2}$ = 0.989; *p* = 0.000), α-amylase ($r^{2}$ = 0.549; *p* = 0.000), cortisol ($r^{2}$ = 0.342; *p* = 0.017) and mBDNF ($r^{2}$ = 0.496; *p* = 0.000). Direct bilirubin showed a positive correlation with indirect bilirubin ($r^{2}$ = 0.858; *p* = 0.000), α-amylase ($r^{2}$ = 0.347; *p* = 0.014), cortisol ($r^{2}$ = 0.486; *p* = 0.000) and mBDNF ($r^{2}$ = 0.623; *p* = 0.000), while indirect bilirubin was positively correlated with α-amylase ($r^{2}$ = 0.600; *p* = 0.000) and mBDNF ($r^{2}$ = 0.429; *p* = 0.001), and nearly significantly correlated with cortisol. Cortisol also showed a positive correlation with mBDNF ($r^{2}$ = 0.608; *p* = 0.000). 

The previous significant correlations between biomarkers and the various questionnaire scores identified at visit 2 did not exist at visit 3, with only near significant correlations between albumin and the DASS21 Stress score and MHC Social Well-being dimension; between total cholesterol and the MHC Psychological Well-being dimension; between glucose and the DASS21 Stress score and the MHC Social Well-being dimension; and between direct bilirubin and the DASS21 Depression Score and Stress score (Figure 9C).

Among the questionnaires at visit 3, the DASS21 Depression score showed a positive correlation with the Anxiety ($r^{2}$ = 0.400; *p* = 0.002) and Stress scores ($r^{2}$ = 0.714; *p* = 0.000) within the same questionnaire, but also a negative correlation with the MHC Emotional Well-being ($r^{2}$ = −0.346; *p* = 0.008), Social Well-being ($r^{2}$ = −0.319; *p* = 0.016) and Psychological Well-being ($r^{2}$ = −0.305; *p* = 0.021) dimensions, and a positive correlation with Quality of Sleep in the LSE ($r^{2}$ = 0.444; *p* = 0.001). The DASS21 Anxiety and Stress scores were only otherwise positively correlated with the Quality of Sleep scores in the LSE (Anxiety: $r^{2}$ = 0.338, *p* = 0.001; Stress: $r^{2}$ = 0.382, *p* = 0.010). Contrary to the associations in visit 1 and visit 2, the PSS Total score at visit 3 was positively correlated with the MHC Emotional Well-being ($r^{2}$ = 0.418; *p* = 0.001), Social Well-being ($r^{2}$ = 0.342; *p* = 0.009) and Psychological Well-being ($r^{2}$ = 0.368; *p* = 0.005) dimensions, as well as with the POMS Total Mood Disturbance scores ($r^{2}$ = 0.348; *p* = 0.009). The MHC dimensions positively correlated with one another (Emotional Well-being and Social Well-being: $r^{2}$ = 0.905, *p* = 0.000; Emotional Well-being and Psychological Well-being: $r^{2}$ = 0.942, *p* = 0.000; Social Well-being and Psychological Well-being: $r^{2}$ = 0.904, *p* = 0.000). The MHC Emotional Well-being dimension was negatively correlated with the POMS Total Mood Disturbance score ($r^{2}$ = −0.343; *p* = 0.010), as were the MHC Social Well-being ($r^{2}$ = −0.386; *p* = 0.003) and Psychological Well-being ($r^{2}$ = −0.376; *p* = 0.004) dimensions. Finally, the Quality of Life score in the AQoL-8D were positively correlated with all three dimensions of the MHC (Emotional Well-being: $r^{2}$ = 0.430, *p* = 0.001; Social Well-being: $r^{2}$ = 0.324, *p* = 0.014; Psychological Well-being: $r^{2}$ = 0.410, *p* = 0.002).

### 3.5. Gut Microbiota

Since relationships were identified between aspects of mental health and biochemical markers, the impact of academic stress and walnut consumption on the gut microbiota was also investigated.

The diversity and composition of the gut microbiota in participants was evaluated in faecal samples. The species richness and evenness of the gut microbiota within samples across the groups was measured using the Shannon index for α-diversity, separately in males and females. In females, a significant main effect of both time (*p* = 0.0024) and treatment (*p* = 0.0123) was evident as well as an interaction effect between time and treatment (*p* = 0.0073; Figure 10). Females in the control group had a lower Shannon diversity index at visit 1 compared to visit 3 (−0.4674 ± 0.13; *p* = 0.0175) and at visit 2 compared to visit 3 (−0.5449 ± 0.13; *p* = 0.0053), while those consuming walnuts showed no significant differences in this metric across all clinical visits, suggesting that walnut consumption was sufficient to stabilise the richness and evenness of the gut microbiota despite the stress experienced during the university semester, particularly at visit 1 and 2. A significant difference in the Shannon diversity index also existed between the control group and treatment group at visit 2 (−0.8735 ± 0.29; *p* = 0.0208). Neither the control group nor the treatment group showed any differences in the Shannon diversity index in males (Appendix A), although this was likely underpowered due to the low recruitment rate of male participants in this study. Thus, academic stress appears to reduce the richness and evenness of the gut microbiota in females, while walnut consumption may attenuate these effects. 

The between-sample dissimilarities in microbial populations were evaluated using the Bray–Curtis β-Diversity index in males and females separately. In females, pairwise PERMANOVA comparisons (Table 3) revealed a significant dissimilarity between the control group at visit 1 and the treatment group at visit 2 (*p* = 0.028), while a near significant dissimilarity was evident in controls at visit 2 versus the treatment group at visit 2 (*p* = 0.083), suggesting that the microbial communities differed in females who consumed walnuts, compared with those that did not. This feature did not appear to be evident in males and pairwise PERMANOVA comparisons (Appendix A) showed that a significant dissimilarity existed in the controls compared with the treatment group at visit 1, however these results need further confirmation due to the low numbers of male participants. Thus, the diversity of the female gut microbiota appears to be influenced by both stress and walnut consumption. Future higher-powered studies are needed to evaluate the effects of academic stress and walnut consumption in males.

Following analysis of the diversity of the gut microbiota in participants, the relative abundances of dominant bacterial phyla were investigated to identify any major shifts in the microbiota. Principle Coordinate analysis of Bray–Curtis dissimilarity as well as taxonomic bar plots in both males and females are available as Appendix A. Since the diversity of the gut microbiota in male participants was largely unaffected in this study, only females were investigated further. No significant differences in Firmicutes, Bacteroidetes, Actinobacteria, Proteobacteria or Verrucomicrobia were identified, however the relative abundance of Actinobacteria in female participants showed opposite trends in controls compared with the treatment group across the 3 clinical visits (Figure 11).

Further analysis of the relative abundance of bacterial families was carried out to investigate the trends identified at the phylum level. No significant differences were found in the most dominant bacterial families (Figure 12), however the trend previously identified for the Actinobacteria phylum was observed in the family *Bifidobacteriaceae*. *Ruminococcaceae* showed a near significant main effect of treatment, however this was most likely due to differences observed at baseline (visit 1) rather than from walnut treatment itself.

Since no major shifts in the abundance of bacterial phyla and families were identified, but significant differences in diversity metrics were evident, we sought to identify changes in the composition of the gut microbiota using the ANCOM in QIIME 2 (2022.2). At the genus level, no significant changes in relative abundance were identified in the commonly reported genera: *Bifidobacteria*, *Bacteroides* and *Lactobacillus* (Figure 13). The ANCOM revealed OTUs of interest which were further investigated by relative abundance analysis. These represented *Ruminococcus 1*, *Ruminococcus 2* and *Alistipes*. In the female samples analysed, *Ruminococcus 1* showed a significant difference in relative abundance overall between visit 1 and visit 3 (−0.0113 ± 0.00; *p* = 0.0128) as well as in controls between visit 1 and visit 3 (−0.0155 ± 0.01; *p* = 0.0464), while *Ruminococcus 2* showed a similar insignificant trend, but only in the treatment group. Since visit 1 could be considered stressful for participants, being held during the first half of the semester where some students had university assessments, the change in relative abundance in *Ruminococcus* between visit 1 and 3 may be due to the cessation of the stressful period at visit 3. Interestingly, *Alistipes* was significantly increased between visit 2 and visit 3 overall (−0.0039 ± 0.00; *p* = 0.0137) with a significant main effect of time on the abundance (*p* = 0.0384). In the control group, there was a significant increase between visit 1 and visit 3 (−0.0042 ± 0.00; *p* = 0.0220) and between visit 2 and visit 3 (−0.0042 ± 0.00; *p* = 0.0068), while remaining unchanged in those consuming walnuts, suggesting that academic stress experienced at visit 1 and visit 2 lowered the abundance of *Alistipes*, but this effect was attenuated at visit 3 when participants had completed the semester.

Finally, the relationships between changes in the gut microbiota, biochemical markers and questionnaires were investigated where similar trends across the clinical visits were identified. Changes identified in the relative abundance of Actinobacteria appeared to follow a similar trend to salivary α-amylase across the three clinical visits. Simple linear regression analyses showed no significant relationship between Actinobacteria and α-amylase levels (Appendix A), suggesting that the changes in these two parameters across the clinical visits may not have been related to one another.

Similarly, changes identified in the relative abundance of *Ruminococcus 1* appeared to follow similar trends to serum mBDNF and the Psychological Wellbeing scores in the MHC questionnaire across the three clinical visits. Simple linear regression analyses showed no significant relationships between *Ruminococcus 1* and serum mBDNF levels, and between *Ruminococcus 1* relative abundance and Psychological Wellbeing scores in the MHC questionnaire (Appendix A). Although these parameters followed similar trends, they do not appear to affect one another.

## 4. Discussion

Here, we show, using a randomised clinical trial, that academic stress in undergraduate university students had a negative impact on overall mental health, exacerbating self-reported levels of stress and depression, and resulting in significant mood disturbances, although there was no evidence of it impacting sleep. Academic stress also disturbed biomarkers of general health and wellbeing, which may be associated with poor mental health. Meanwhile, walnut supplementation for 16 weeks in undergraduate university students had a positive effect on self-reported levels of stress and depression, improved overall mental health, and aided sleep in the longer term. However, walnut supplementation was not sufficient to influence mood as measured by the Profile of Mood States (POMS) questionnaire. Furthermore, we show associations between perceived stress, mental health and sleep quality with biomarkers of general health and well-being. Lastly, we report on the effects of academic stress and daily walnut consumption on the diversity and composition of the gut microbiota.

### 4.1. Questionnaires

Pribis [17] previously found no effect of walnut consumption on mood in university students in a crossover designed clinical trial using the POMS questionnaire, although their trial took place prior to the university examinations to avoid the effects of stress. Here, we show similar findings before, but also during and following, the university examination period since no interaction effects of walnut consumption and time were found in any of the 5 dimensions in the POMS. However, academic stress was sufficient to cause an increase in total mood disturbance overall. Of note are the dimensions for Anger-Hostility and Depression-Dejection which both showed trends indicating walnuts stabilised these across the clinical visits, although this did not reach significance. Interestingly, participants who consumed walnuts throughout the study had a near significant decrease in Depression-Dejection scoring at the final visit compared with visit 1, with an overall stable trend across the clinical visits compared to controls, suggesting that walnuts may have influenced emotions related to depression or dejection.

The DASS21 similarly showed that academic stress was sufficient to increase the negative emotional states of depression in participants, since the scores on this subscale were significantly increased during visit 2, coinciding with the university examination period. The near significant interaction effect between walnut consumption and time may indicate that walnuts reduce negative emotional states of depression which may arise from academic stress. Walnut consumption has been associated with a decreased risk of developing depression [15]. A recent systematic review corroborated this by concluding that higher nut consumption may be associated with fewer depressive symptoms and enhanced mood states [13]. We found similar results for negative emotional states of anxiety in participants using the DASS21, with a significant interaction effect between walnuts and time, but no significant differences in scores between each clinical visit in participants who consumed walnuts, suggesting that walnuts stabilised negative emotional states of anxiety even during periods of academic stress. Negative emotional states of stress, as measured by the DASS21 subscale, were significantly increased by academic stress at visit 2, however participants who consumed walnuts showed a trend suggestive of walnuts stabilising these negative emotional states too. 

The PSS supported these results indicating that the examination period was sufficient to induce negative emotions related to perceived stress, however walnut consumption again showed trends indicative of the stabilisation of negative emotions related to stress, since no significant differences existed in this subscale across the clinical visits in participants who consumed walnuts. Taken together with results from the DASS21, this suggests that walnuts may have a positive effect on stress, anxiety and depression experienced by university students. These effects of walnut supplementation may be due to the omega-3 fatty acid content of walnuts since evidence suggests omega-3 supplementation may alleviate anxiety and depression [39,40]. An analysis of 26,656 participants in the National Health and Nutrition Examination Survey (NHANES) revealed that depression occurred less in individuals who consumed walnuts compared with non-nut consumers, with this effect being strongest in women [15]. However, a scoping review recently concluded that omega-3 consumption in individuals aged 14–24 could not improve symptoms of anxiety and depression [41]. Parilli-Moser et al. [42] showed in their randomised controlled trial that peanut consumption for 6 months could lower measures of depression and anxiety in young adults, and this was attributed to polyphenols present in peanuts altering faecal short-chain fatty acid levels. Thus, the positive effects of walnut consumption in our study may not be due to the omega-3 fatty acid content of walnuts nor the polyphenol content since walnut consumption in our study was unable to affect plasma polyphenol levels (see Section 4.2 to follow), warranting further studies to elucidate the nutritional component of walnuts which may lead to these effects.

The positive effects of walnut consumption on stress, anxiety and depression were synonymous with self-reported levels of mental well-being since the MHC showed that academic stress had a negative impact on the psychological well-being subscale, although participants who consumed walnuts throughout the study had no significant differences in scores for this measure across all clinical visits, suggesting a protective effect of walnuts on mental health. Interestingly, this was consistent with the Mental Health dimension of the AQoL-8D, since a near significant interaction effect was observed in participants who consumed walnuts, which showed walnuts prevented a decline in mental health during the university examination period. Many studies have highlighted the growing problem of poor mental health in undergraduate students and a recent World Health Organisation Mental Health Survey found at least 75% of mental health disorders occur prior to 24 years of age [2,3,4]. Furthermore, a recent meta-analysis has also suggested that sleep is causally related to challenges in mental health with improvements in sleep reducing depression, anxiety and stress [43]. 

In university students, poor sleep quality has been associated with a reduction in mental health, regardless of generally good sleeping habits [44]. Walnut consumption has also been shown to increase blood melatonin levels in rats which plays a vital role in brain function and sleep [45,46]. Yeh et al. [46] previously demonstrated walnut oligopeptides were able to improve memory, cognition and sleep quality in teenagers and the elderly, likely due to the regulation of melatonin levels. While melatonin levels were not measured in this study, the existence of sleep disorders were evaluated in participants using the LSE. For the four dimensions evaluating ease of getting to sleep, sleep quality, awakening from sleep and behaviour following wakefulness, the university examination period during visit 2 had no effect on any of these sleep measures, however, walnut consumption showed a significant improvement in all four measures at visit 3, suggesting that walnuts may improve general sleep quality in the longer term. 

Correlation analyses of the questionnaires revealed that measures of stress such as the Total Score in the PSS shared a relationship with measures of mental health in the MHC. At visit 1, PSS Total Score showed a mild negative correlation with all three dimensions of the MHC, suggesting that higher levels of stress reduced mental well-being, while at visit 3 this relationship was positively correlated, coinciding with lower PSS scores at visit 3. These results suggest, firstly, that increased stress negatively impacts mental health and well-being, and secondly, although visit 1 was considered a “baseline”, participants had already commenced the university semester and may have been experiencing stress due to assessments in the first half of the semester. Although, the DASS21 Stress Score showed no significant correlations with any of the MHC dimensions at all three clinical visits. In fact, the DASS21 Stress Score did not correlate with the PSS Total Score at any of the clinical visits, however this may be due to DASS21 being validated to measure depression rather than stress directly [47]. The PSS has been validated as a stress measure in university students [48]. Though the DASS21 has had its internal validity widely confirmed, its subscales may be insufficient to measure anxiety and stress individually since a recent study found that the DASS21 was unable to distinguish patients diagnosed with anxiety or depression from each other [47,49]. Regarding depressive emotionality, the DASS21 Depression Score showed a negative correlation with all three dimensions of the MHC only at visit 3, suggesting that levels of perceived depressive emotionality experienced at visit 1 and visit 2 may have contributed to the gradual decline in mental health and well-being across the three clinical visits. Lastly, Overall Quality of Life in the AQoL-8D, which shared trends with its subdimensions of mental health and coping, showed a positive correlation at all three clinical visits with the MHC, suggesting that overall quality of life is influenced by mental health. Of interest to this study was the participants’ reported general mood, emotionality and mental health, and the relationship with biochemical markers of general health.

### 4.2. Biomarkers of General Health and Well-Being

Walnuts are high in protein, minerals, fatty acids and phytochemicals such as phenolic compounds and tocopherols and have long been touted for their wide-ranging health benefits. In particular, recent research suggests that extracts of walnuts may induce increased antioxidant activity providing protection against ageing and cancer [50]. A recent systematic review found diets enriched with walnuts may decrease biomarkers of metabolic syndrome, inflammation and predictors of senescence and age-related diseases [51]. In our study, walnut supplementation did not increase the total polyphenol content or ferric reducing ability and oxygen radical absorbance capacity of plasma in the study participants. Previous studies have suggested that consumption of walnuts results in a significant, but acute increase in plasma polyphenol content and antioxidant capacity, which has been shown to significantly decrease within 150 min of consumption [52]. Participants in the present study were not instructed to consume walnuts at a predetermined timepoint relative to their clinical visits, which may have resulted in lack of the effects of walnuts on these parameters. Furthermore, studies which evaluated the long-term effects of walnut consumption have shown that these parameters were unaltered, suggesting that the acute effects of walnut consumption may not cumulate with long-term consumption [53,54]. Thus, our results are consistent with previous studies. Academic stress in the present study had no effect on these measures. Sivoňová et al. [55] showed that academic stress could reduce plasma antioxidant capacity in university students, however in their study participants were instructed to avoid excessive intake of antioxidant vitamins (or equivalent natural products), juices, fruits and vegetables, which was not a requirement for participants in our study.

This study also investigated the effects of walnut consumption on biomarkers of general health and well-being. While serum glucose and triglyceride levels were unaffected throughout the university semester or by walnut consumption, cholesterol levels were significantly decreased (Appendix A) at visit 2 compared with baseline, suggesting that academic stress may lower serum cholesterol levels, and this may be related to the increased DASS21 Depression scores at visit 2. A negative correlation was found between DASS21 Depression scores and total cholesterol levels at visit 1 and visit 2, but this association was not present at visit 3 (Figure 9), suggesting that depressive emotionality may be associated with decreased total cholesterol. According to Zwolińska et al. [56], recent studies have drawn conflicting conclusions showing both increased and decreased total cholesterol in depression. This is in contrast to a recent umbrella meta-analysis suggesting that decreased total cholesterol levels are highly indicative of depression [57]. Zwolińska et al. [56] suggested that increased cholesterol levels may upregulate the inflammatory response, leading to depression pathophysiology through increased proinflammatory cytokines, IL-6 and TNF-α. However, since depressed individuals present with lower high-density lipoprotein (HDL) and higher low-density lipoprotein levels (LDL), it may be that a higher LDL/HDL ratio leads to immunological activation rather than increased total cholesterol levels [58]. Lowered serum levels of cholesterol may be related to the development of depression, even though cholesterol cannot cross the blood–brain barrier, oxysterol metabolites of cholesterol such as 24- and 27-hydroxycholesterols can cross the blood–brain barrier and increase inflammation. Increased levels are also detected in the brains of suicides [59,60]. Thus, the lowered levels of serum cholesterol found in this study may have contributed to the negative emotionality states reported at visit 2.

Biomarkers of kidney function measured in this study, namely serum urea and creatinine, did not indicate that kidney function was altered by academic stress or walnut consumption, since an increase in serum urea in controls occurred at visit 3 compared with visit 1 and visit 2, but this was not accompanied by changes in creatinine levels (Appendix A). The increase in serum urea in controls at visit 3 may be related to the increased protein metabolism by the liver, and/or mild dehydration, possibly as a result of chronic stress, rather than altered kidney function. While biomarkers of liver function did not indicate alterations in general liver function (Appendix A), serum total protein levels were significantly decreased at visit 3 compared with visit 1 in controls (Appendix A). A poorer diet later in the semester, particularly during the examination period, may have contributed to these lowered levels, although self-reported dietary habits were not altered throughout the clinical visits. This decrease in serum total protein levels did not occur in participants who consumed walnuts. In fact, walnut consumption significantly increased total protein levels at visit 2 with this effect sustained at visit 3 in comparison with the controls (Table 2). This effect was similarly reflected in serum albumin levels with walnut consumption increasing levels relative to controls at both visit 2 and visit 3 (Table 2). Given albumin contributes to 60% of total protein levels [61], together these results suggest that the incorporation of walnuts in the diet may increase albumin levels. This may be of importance since lowered albumin levels have been independently associated with both malnutrition and inflammation [62]. Furthermore, decreased albumin levels have been associated with increased presence and severity of depression, possibly by contributing to a reduction in peripheral tryptophan levels and serotonin synthesis since albumin tightly binds tryptophan, a precursor of serotonin [63,64]. Thus, the decreased total protein levels in the control group at visit 2 and visit 3 may have been partly influenced by decreased albumin levels. While albumin was not significantly correlated with any measures in the DASS21 or PSS, total protein levels had a near significant negative correlation with DASS21 Stress scores at both visit 1 and visit 2, but not visit 3 (Figure 9). A significant negative correlation also existed with both the Emotional Well-being dimension of the MHC and Overall quality of Life in the AQoL8D at visit 2, but not at visit 1 and visit 3, while albumin followed the same trend (Figure 9). 

Regarding the biomarkers of stress measured in this study, both cortisol and α-amylase appeared to be unaffected by examination stress experienced by participants at visit 2 (Figure 7), even though participants in the control group experienced significantly increased perceived stress based on the DASS21 and PSS. Moreover, α-amylase and cortisol showed a weak but insignificant negative association to each other at visit 1 and visit 2, but not visit 3 (Figure 9), despite Chaturvedi et al. [65] showing a positive correlation in their study. This may have been a limitation of both studies since samples were collected between 12pm and 4pm in our study, compared to between 10 a.m. and 1 p.m. [65]. Salivary α-amylase is known to have a distinct circadian pattern being low in the morning and steadily increasing throughout the day, whereas cortisol is known to be highest after awakening and steadily decreases throughout the day [66,67]. It may also be that the HPA axis response in participants in our study may have been too mild to result in significant changes during the examination period since Preuß et al. [68] previously showed a mild increase in cortisol levels in university students undertaking written examinations compared to a strong increase for those undertaking oral presentations and the majority of participants in our study had written examinations. Interestingly, our study found a significant moderate negative correlation between cortisol levels and bilirubin levels at visit 1 and visit 2, but a significant positive correlation at visit 3 (Figure 9), although bilirubin levels were within a healthy range. This relationship may warrant further study since cortisol levels have been negatively associated with chronic liver disease and depression is common in such patients [69,70]. Regarding α-amylase, a negative correlation with both total protein and total cholesterol was evident at visit 2, but not at visit 1 and visit 3 (Figure 9). Previous studies have found that salivary α-amylase is higher in depression, decreases during antidepressant treatment and may have potential as a biomarker of depression, sharing a negative correlation with cholesterol levels [6,71]. Considering depressive emotionality was higher in participants at visit 2 compared with the other clinical visits as indicated by the DASS21, particularly in controls, the relationship between salivary α-amylase and serum total cholesterol in our study was similar to that seen in depressive patients, indicating that these biomarkers may have relevance even in mild depression and chronic stress. Interestingly, participants who consumed walnuts had significantly lower α-amylase levels at visit 2 compared with controls (Figure 7), although the same effect did not occur with total cholesterol levels, suggesting that the positive effects of walnut consumption on mental health may also be reflected in the biochemistry of stress.

Serum mBDNF was evaluated in this study as an indication of brain health. Mitoma et al. [72] previously identified a mild negative correlation between perceived stress and serum mBDNF levels in healthy individuals, suggesting that stress may affect peripheral mBDNF levels. We found serum mBDNF levels were decreased in controls at visit 2 and visit 3, while participants who consumed walnuts followed a similar trend, but only had significantly decreased levels at visit 3 (Figure 8). These results suggest that academic stress may have a long-term effect on serum mBDNF levels since mBDNF levels remained lower during visit 3 even after cessation of stress. Although mBDNF levels did not correlate with any of the questionnaire scores related to depressive emotionality, they possibly could be associated with impaired cognition and memory because of prolonged periods of stress which were not measured in this study. Indeed, serum mBDNF has positively correlated with cognitive performance in humans [73,74]. Though serum mBDNF levels are thought to reflect mBDNF levels in the brain since depressive patients exhibit decreased serum mBDNF levels, the relationship between these two pools of mBDNF is poorly understood since a body of research has also found that peripheral mBDNF levels are related to platelet activation and that platelet abnormalities occur in depressive patients [75,76,77]. Some studies have also found a positive correlation between serum mBDNF levels and serotonin, the major neurotransmitter thought to be dysregulated in depression, although these results have not been reproduced [78]. However, it is well established that mBDNF in the brain increases the plasticity of serotonergic neurons, and thus is implicated in depression pathophysiology [78,79]. More recently, a pilot study identified a positive correlation between serum mBDNF and grey matter parameters in multiple depression-related brain regions in patients with mild, but not major depression, suggesting that early depressive states may be reflected in serum mBDNF [80]. Another study found no relationship between serum mBDNF and hippocampal volume in healthy individuals from a broad age range [81]. In the present study, while physical parameters of the brain were not measured, negative emotionality related to stress, anxiety and depression was measured using the questionnaires previously discussed. No significant correlations were found between serum mBDNF levels and any of the questionnaire scores throughout our study (Figure 9), thus the relation of the decreased levels of serum mBDNF identified in this study is not clear. Future studies should evaluate serum mBDNF levels in conjunction with cognition and memory to better elucidate the relationship between peripheral mBDNF and implications of stress.

### 4.3. Gut Microbiota

It is currently established that dysbiosis of the gut microbiota occurs in psychiatric disorders such as major depression, however the effects of chronic mild stress, such as that experienced by university students, on the diversity and composition of the gut microbiota is less clear [82,83]. Initial analysis of the gut microbiota in both male and female participants combined revealed no significant changes to diversity metrics nor composition (results not shown). Taking sex into account revealed possible effects of sex on the diversity of the gut microbiota since females in the control group had lower diversity metrics during the university semester (visit 1 and visit 2) compared to levels during the university break (visit 3), while diversity metrics in male participants in the control group remained unchanged throughout the study, although the results from male participants are likely underpowered due to the low number of participants relative to females (Appendix A). The possible effects of sex and stress on the gut microbiota phenotype may be attributed to crosstalk between sex hormones and the gut microbiota, since oestradiol and progesterone are known to contribute to the higher vulnerability of developing anxiety and depression disorders in women [84,85]. Regarding the gut microbiota composition, which was only measured in females, both the genera *Ruminococcus 1* and *Alistipes* were identified to have higher abundance during the university break (visit 3) rather than during the semester (visit 1 and visit 2) (Figure 13). Previous studies have found that stress affects *Ruminococcus* abundance, although preclinical rodent studies show *Ruminococcus* abundance is negatively correlated with stress, while *Ruminococcus* abundance in female humans appears to increase with increased stress [86,87]. *Alistipes*, a relatively newly identified bacterial genus, shows increased abundance in depression [88,89]. Interestingly, both *Ruminococcus* and *Alistipes* may be involved in the regulation of neurophysiology of the gut-to-brain axis by influencing tryptophan and serotonin levels in the gut. According to Kaur et al. [90] and Parker et al. [89], *Ruminococcus* and *Alistipes* both produce tryptamine from dietary tryptophan, thus they could prevent the production of colonic serotonin, influencing host intestinal physiology, but they may also in this way prevent absorption of dietary tryptophan into the host’s bloodstream, in turn influencing serotonin levels in the brain since tryptophan, not serotonin, is able to cross the blood–brain barrier [91]. While serotonin levels were not measured in this study, scores from the MHC indicate participants in the control group experienced poorer psychological well-being at visit 2 and 3, thus the increased relative abundance of *Ruminococcus* at visit 2 and 3, and *Alistipes* at visit 3 may have contributed to these effects. Contrarily, *Rumminococcus* has also been considered a probiotic and there is evidence that it is decreased in major depression, thus an alternative explanation to the lower *Ruminococcus* levels at visit 1 compared with visit 3 may be that participants experienced more stress at visit 1, rather than at visit 2, since the first half of the semester would have included several university assessments and the written examinations at visit 2 may have induced a milder HPA axis response in comparison [68]. The HPA axis response is also thought to influence intestinal homeostasis, similarly to the brain, via corticotropin-releasing factor receptors expressed in the gut which increase intestinal permeability and local inflammation, in turn modulating the gut microbiota [92].

In participants who consumed walnuts throughout the study, diversity metrics related to richness and evenness were unchanged in both males and females (See Appendix A), suggesting that walnuts stabilised the diversity of the gut microbiota throughout periods of stress, remaining higher overall compared with controls, at least in females (Figure 10). β-Diversity metrics showed that walnut consumption resulted in a near significant difference in overall composition of the gut microbiota compared with controls, in females. (Table 3) Compositionally, at the genus level, both *Ruminococcus* and *Alistipes* remained unchanged across all 3 clinical visits in females who consumed walnuts (Figure 13), suggesting that walnut consumption may have counteracted the effects of stress on the relative abundance of these genera, at least in females. It is well known that sex hormones modulate the enzymatic conversion of polyunsaturated fatty acids into long-chain polyunsaturated fatty acids such as the beneficial omega-3 fatty acids, particularly in females [93], and since walnuts contain high amounts of omega-3 fatty acids, the different comparative trends in males and females in the control versus the treatment groups suggest that walnut consumption may have benefitted the gut microbiota in females. Furthermore, the relative abundance of the butyrate producing genus, *Ruminococcus*, remained higher and more stable relative to controls in females. This higher overall relative abundance of *Ruminococcus* may be related to how consumption of omega-3 fatty acids affects the gut microbiota since previous studies have found associations between its consumption and butyrate levels in the gut, which may be suggestive of increased butyrate production by the gut microbiota [94]. The relative abundance of *Alistipes* in female participants who consumed walnuts versus their control counterparts remains an interesting result since walnuts appear to have prevented an increase in the abundance of this genus as a result of academic stress. *Alistipes* has previously been implicated to improve disease states such as colitis and fibrotic disorders, but also act as a pathogen in other disorders such as chronic fatigue syndrome, depression and anxiety [89]. Thus, the long-term consumption of walnuts may counteract the effects of stress and depression on the gut microbiota in females. Previous studies have separately shown that antidepressant treatment causes a reduction in *Ruminococcus* abundance and an increase in mBDNF levels in the brain, although not all types of antidepressants reflect these changes in mBDNF levels in serum [95,96]. To our knowledge, no studies have reported on any potential relationship between *Ruminococcus* abundance and serum mBDNF levels. Our study showed that the identified changes in the female gut microbiota shared no significant relationships with any of the biochemical parameters or questionnaire scores affected by academic stress in this study. Future higher-powered studies are needed to analyse potential sex-dependent effects of academic stress on the gut microbiota, since our study had far fewer male participants than females, and to further elucidate how changes in biochemical parameters are related to the changes in the gut microbiota in chronic stress.

### 4.4. Limitations of the Study

This study had limitations in that, apart from the gut microbiota, results were analysed without accounting for potential sex differences, thus some results may have been better correlated had sex been accounted for in all analyses. Unfortunately, due to the low numbers of male participants in our study it was not possible for us to establish the effects of walnuts or academic stress in males and thus, sex-dependent mechanisms require further study. A further limitation was that the effects of daily walnut consumption identified may have been a placebo effect since participants were not blinded to treatment, thus there may have been anticipatory effects. The effects of both academic stress and walnut consumption may have also been confounded by the 2020 COVID-19 pandemic since cohort 2 attended clinical visits during this period, which were also disrupted by a brief stay-at-home order by the Australian government. Furthermore, the nature of the university examinations for cohort 2 and cohort 3 differed to those in cohort 1 since the University of South Australia moved from in-person to online university examinations in response to the COVID-19 pandemic.

## 5. Conclusions

Through the randomised clinical trial used in this study, we have shown that academic stress in undergraduate university students has a negative effect on mood and overall mental health. Key findings are summarised in Box 1. 

Box 1The effects of academic stress and walnut consumption on mental health and general well-being in university students.Academic stress had a negative impact on mental health as evidenced by decreased scores for Mental Health and Psychological Well-being in the AQoL-8D and MHC, respectively.Academic stress increased self-reported levels of stress
and depression (DASS21) during the university examination period.Daily consumption of walnuts prevented the significant
changes in mental health-related scores (AQoL-8D & MHC) and scores of
stress and depression (DASS21), thus walnuts may alleviate the negative
effects of academic stress on mental health in university students.Academic stress had a negative effect on metabolic
biomarkers such as total protein and albumin since these were decreased
during the university examination period.Daily consumption of walnuts increased total protein and
albumin levels, thus may protect against the negative effects of academic
stress on metabolic biomarkers.While academic stress did not change stress biomarkers
such as cortisol and α-amylase, daily walnut
consumption decreased α-amylase levels, further suggesting that walnuts may
protect against the effects of stress.Academic stress was associated with lower gut microbial
diversity in females.Daily walnut consumption may alleviate the negative
effects of academic stress on the diversity of the gut microbiota in females.Walnut consumption may improve sleep in the longer term.

While daily consumption of walnuts could not alleviate disturbances in mood, it had a protective effect against the negative impacts of academic stress on mental health. These protective effects were reflected in relevant biomarkers of stress such as salivary α-amylase. Academic stress may also have a negative effect on the diversity of the gut microbiota in females, however further studies are needed to confirm this effect in males. Interestingly, daily walnut consumption over 16 weeks was able to alleviate the negative effects of academic stress on the diversity of the gut microbiota in females, however the relevance of these changes to the biochemistry of chronic stressors such as academic stress requires further study.

## Data Availability

Not applicable.

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
