# Peer review of "The Effects of Walnuts and Academic Stress on Mental Health, General Well-Being and the Gut Microbiota in a Sample of University Students: A Randomised Clinical Trial"

_nutrients, 2022, doi:10.3390/nu14224776_

Round 1

Reviewer 1 Report

In this study, authors evaluated the relationship between academic stress, mental health, biochemical markers (total protein, albumin, cholesterol…) of general health, and the diversity and composition of the gut microbiota. And authors evaluated the effects of daily walnut consumption on these parameters in a sample of university students.

Authors showed that academic stress in undergraduate university students has a negative effect on mood and overall mental health. Academic stress is sufficient to negatively impact metabolic and stress biomarkers. Although daily consumption of walnuts did not alleviate disturbances in mood, it had a protective effect against the negative impacts of academic stress on mental health. Authors also found that daily walnut consumption over 16 weeks was able to alleviate the negative effects of academic stress on the diversity of the gut microbiota in females.

This study was designed and conducted very well. The findings are important, and conclusions are convincible. Only minor revision is needed for the last few paragraphs:

1) the Line 1077-1085 are that same as line 1099-1105.

2) Please indicate the p value in table 2&3, which groups are compared.

Reviewer 2 Report

This randomized clinical trial provided evidence that, in undergraduate university students, academic stress negatively impacted mood and overall mental health as well as metabolic and stress biomarkers. However, consumption of walnuts showed protective effects and, in female participants, improvement of microbiota affected by academic stress.

The topic is important and the study provides a comprehensive analysis of the subject. I would suggest the following questions:

- what were the reasons you decided on two servings (56 g) of walnuts per day and 16 week duration?

- acronyms/abbreviations should be used after they were defined 

- Scheme 2 should be included in text

- please check that all Tables and Figures (including those in Supplementary Materials) are mentioned in the text

- please check the math in Scheme 2 (Treatment=41 and Control=39 do not add up to Randomised=82) 

Line 86: is it “2.2 Study Design”? Same Line 205

Line 139” “were determined”

Line 178: “data... were further ...” 

Line 204 (Results): first paragraph and Table 1 should go to Materials and Methods

Line 399: Table 2 can be moved to Supplementary Materials

Line 442: “Figure 7” instead of “Figure 8”

Line 461: “Figure 8” instead of “Figure 9

Line 855: The following important findings can be added here - “Biomarkers of metabolic syndrome and inflammation, predictors of senescence and age-related diseases, could be decreased by walnut-enriched diets with no consequences on anthropometric and glycemic parameters (doi: 10.3390/antiox11071412)”.
